# Screening of phytoconstituents from *Bacopa monnieri (L.) Pennell* and *Mucuna pruriens (L.) DC.* to identify potential inhibitors against *Cerebroside sulfotransferase*

**Nivedita Singh**⊙*, **Anil Kumar Singh**

Department of Dravyaguna, Faculty of Ayurveda, Institute of Medical Sciences, Banaras Hindu University, Varanasi, Uttar Pradesh, India

* niv234@gmail.com, s.nivedita@bhu.ac.in

**Data Availability Statement:** All relevant data are within the paper and its Supporting Information files.

## Abstract

Cerebroside sulfotransferase (CST) is considered a target protein in developing substrate reduction therapy for metachromatic leukodystrophy. This study employed a multistep virtual screening approach for getting a specific and potent inhibitor against CST from 35 phytoconstituents of *Bacopa monnieri (L.) Pennell* and 31 phytoconstituents of *Mucuna pruriens (L.) DC.* from the IMPPAT 2.0 database. Using a binding score cutoff of -8.0 kcal/mol with ADME and toxicity screening, four phytoconstituents IMPHY009537 (Stigmastenol), IMPHY004141 (alpha-Amyrenyl acetate), IMPHY014836 (beta-Sitosterol), and IMPHY001534 (jujubogenin) were considered for in-depth analysis. In the binding pocket of CST, the major amino acid residues that decide the orientation and interaction of compounds are Lys85, His84, His141, Phe170, Tyr176, and Phe177. The molecular dynamics simulation with a 100ns time span further validated the stability and rigidity of the docked complexes of the four hits by exploring the structural deviation and compactness, hydrogen bond interaction, solvent accessible surface area, principal component analysis, and free energy landscape analysis. Stigmastenol from *Bacopa monnieri* with no potential cross targets was found to be the most potent and selective CST inhibitor followed by alpha-Amyrenyl acetate from *Mucuna pruriens* as the second-best performing inhibitor against CST. Our computational drug screening approach may contribute to the development of oral drugs against metachromatic leukodystrophy.

## 1. Introduction

Metachromatic Leukodystrophy (MLD) is an autosomal recessive neurodegenerative hereditary disease that belongs to the group of lysosomal storage disorders (LSDs) [1–3]. MLD is characterized by progressive demyelination of oligodendrocyte and Schwann cells in the central and peripheral nervous system, which subsequently leads to neurological dysfunction including developmental delay, progressive motor dysfunction, astrocyte dysfunction, rising

**Funding:** Institute of Eminence, Banaras Hindu University, Government of India for providing postdoctoral fellowship and research grant for funding this work through research Grant No.: R/ Dev/G/6031/IoE/MPDFs/61698.

**Competing interests:** The authors declare that there are no competing interests associated with the manuscript.

inflammation, and communication gap, in severe cases, it may lead to death, etc. [2, 4–7]. MLD is caused by the accumulation of sulfatide (3-O-sulfogalactosylceramide) glycolipids and their toxic metabolites with mutation in lysosomal enzyme, *Aryl sulfatase A* (*ARSA*) [5, 7, 8]. Apart from the nervous system, sulfatide accumulation also occurs in the kidneys and gall bladder, hence the overall system is compromised [7]. Worldwide, the prevalence rate of MLD is one in every 40,000–1,60,000 population, making it a rare disease of greater concern [1, 6]. MLD is largely categorized into three main clinical types: late-infantile (<30 months), juvenile (early juvenile [30 months–6 years] and late juvenile [7–16 years], and adult MLD (>17 years) [3, 9–13].

At present, there is no effective and reliable therapy for this devastating disease particularly at the early stage (less than 7 years) for disease management. The existing therapies including gene therapy, enzyme replacement therapy, chaperone therapy, hematopoietic stem cell therapy, etc., are under different stages of clinical trials and are largely ARSA-dependent [6, 11, 14–16]. These treatments vary on a case-by-case basis as more than 280 mutations of the *ARSA* gene have been reported so far, thus, high cost and high risk are associated with the success of these treatments [2, 17]. In recent times, with the development of state-of-the-art technologies, substrate reduction therapy (SRT) is emerging as a potential treatment strategy and alternative to existing therapies focusing on the development of oral drugs via developing inhibitors targeting the enzyme responsible for the biosynthesis of the substrate. Thus, SRT is a rate-limiting strategy to control the accumulation of the substrate of the deficient enzyme by regulating the catalytic activity of the predecessor enzyme [18, 19]. In recent times, SRT has been successful in various single-gene disorders including other lysosomal storage disorders [5, 20–22]. Miglustat and Eliglustat are the two FDA-approved oral drugs for Gaucher's disease [22–24].

In MLD, for substrate reduction therapy, the target enzyme is *Cerebroside sulfotransferase* (CST), which is the final enzyme in sulfatides biosynthesis [2, 5, 12, 25]. CST is a membrane-bound protein, that catalyzes the transfer of the sulfuryl group from the donor co-substrate, PAPS to the acceptor substrate, galactosylceramide (GC) to synthesize sulfatide [25–27]. The development of a competitive inhibitor for the substrate is a strategy to counter the sulfuryl transfer mechanism to regulate the level of sulfatide.

So far, SRT has been in its nascent stage and unable to use the latest advancement in preliminary stage of *in silico*-based drug screening because of the absence of experimentally derived three-dimensional structure of CST protein and lack of availability of structural data [2]. To fill this gap, recently our research group has successfully developed a three-dimensional computational model of CST protein which gives a better understanding of the catalytic action of CST by identifying the critical active site residues for developing specific and potent CST inhibitors [12]. Through CST model development, we have somehow effectively dealt with the issue of the unavailability of CST structure. Another major challenge in developing effective substrate reduction therapy is to find suitable lead molecules to ensure the selectivity and potency of drug candidates towards CST while minimizing the side effects of other proteins. The present study is an effort in this direction to identify potential drug candidates for SRT as well as potential lead molecules for future research.

Various plants are known for enhancing memory and cognitive functioning along with treating various forms of neurodegenerative disease. The source of the majority of our modern neuroprotective medicines can be traced to phytoconstituents [28–32]. *Bacopa monnieri* and *Mucuna pruriens* are two such potential nutraceutical herbs known to be mood and memory enhancers and cognitive function promotors [33–36]. Their unique phytochemical composition makes them potential herbs for screening drug candidates for CST inhibition for MLD. In the present times, bioinformatics has evolved with the availability of large databases of natural

and synthetic compounds, software, web tools, and servers for optimization of structure, screening of small molecules, and pharmacokinetic studies [37–41]. These are cost-effective methodologies for preliminary drug development. The present study is an effort to screen available phytoconstituents of *Bacopa monnieri* and *Mucuna pruriens* using computational approaches to arrest MLD and make them available for future experimental studies.

## 2. Materials and methods

### 2.1 Protein preparation and grid generation

The three-dimensional model of CST prepared by our group using a computational modeling approach was used in this study as a 'receptor' [12]. The model protein is comprised of amino acid residues between 69–336 of the full-length protein sequence of 423 amino acids, covering the entire catalytic region of the protein [2]. The protein catalytic site comprises a substrate (galactocerebroside) binding site and a co-substrate (PAPS) binding site on a linear horizontal plane of the protein where the co-substrate is the sulfuryl donor and the substrate is sulfuryl acceptor [2, 12]. The 500ns simulated protein model proceeded by removing water molecules and heteroatoms and adding hydrogen with proper assignment of atom type and Gastegier charge, to generate a PDBQT file of protein using AutoDock 4.2. Afterward, considering the key substrate binding sites residues Lys82, His84, Lys85, Asn113, His141, Phe170, Tyr176, Phe177, Tyr203, and His212, a grid (.gpf) was generated with a grid box of 90 × 90 × 90 Å and spacing of 0.253 Å.

### 2.2 Ligand preparation

3D (.mol2) structures of 35 phytochemicals from *Bacopa monnieri* and 31 from *Mucuna pruriens* were retrieved from the IMMPAT 2.0 (Indian Medicinal Plants, Phytochemistry And Therapeutics) online database [39, 42]. Using AutoDock Racoon virtual screening file preparation software, the.mol2 files of these compounds were converted to.pdbqt files after energy minimization and assignment of atom types [43].

### 2.3 Molecular docking based virtual screening

Following the preparation of the protein, ligands, and grid files, docking (.dpf) files were prepared in AutoDock Racoon for each ligand. Subsequently, Racoon arranged all these files in separate folders for each ligand along with the generation of a single virtual screening script (.sh) file for molecular docking to run in the Linux for all ligands simultaneously. Molecular docking was performed in the Param Shivay HPC Linux system (837 TFLOPS capacity with Intel(R) Xeon(R) Gold 6148 CPU @ 2.40GHz and 40 CPU per node) using AutoDock 4.2.6 with docking parameters including 100 GA run, 300 population size, 27000 maximum number of generations, and 25000000 maximum number of evaluations along with applying Lamarckian genetic algorithm and a gradient-based local search method. Based on the lowest free energy of binding, the best-docked conformation was selected, and processed using custom Python scripts, and visualized using.pdb visualization tools including PyMol and Discovery Studio visualizer to analyze non-covalent interactions and binding patterns.

### 2.4 In silico drug-likeness properties, ADME, and toxicity analysis

To perform pharmacokinetic properties of top hits (selected based on binding score), pKCSM, SwissADME, and ProTox web servers were employed [44–46]. The canonical SMILE of selected compounds was used as entry data in these systems for providing the pharmacokinetics data for each compound. The pharmacokinetic analysis considered adsorption,

distribution, metabolism, and excretion along with toxicity analysis. Drug likeness properties were analyzed using the Lipinski rule of 5.

## 2.5 MM-PB/GBSA analysis

Protein-ligand docked complexes of compounds selected from ADMET analysis were considered for calculating binding free energy using MM/PB(GB)SA methods using the farPPI webserver using the ff14SB force field for small molecule and the Generalized Amber Force Field2 (GAFF2) for receptor analysis [47, 48]. The binding free energy of the selected compounds was calculated using the molecular mechanics procedures including PB3, PB4, GB1, GB2, GB5, and GB6 for MM/PB(GB)SA calculation. The number followed by PB is Poisson–Boltzmann calculation while GB is the generalized Born calculation. The input ligand and protein files were separated from the docked conformation file obtained from molecular docking. The higher negative MMGBSA value represents the higher degree of rigidity of the ligand-protein complex. The binding free energies were calculated using Eq 1.

$$\Delta G_{bind} = G_{complex} - (G_{receptor} + G_{ligand}) = \Delta H - T\Delta S = \Delta E_{MM} + \Delta_{solv} + T\Delta S \qquad (1)$$

## 2.6 Molecular dynamics (MD) simulations

The Molecular dynamics simulations of the screened protein-ligand complex were performed to optimize and determine the overall stability of the protein in protein-ligand complexes under atomistic simulation conditions [49–51]. The simulation was performed in GROMACS 2023 using Charmm 27 all-atom additive force field. For the simulation, a dodecahedron simulation box was created with a minimum distance of 1.2 Å from the box edge to apply periodic boundary conditions and minimize the edge effect. TIP3P water solvation model was used, and the system was then neutralized by the addition of chloride (Cl⁻) ions and subsequently the system was energy minimized using the steepest descent algorithm. The system was thereafter subjected to 1000ps (1ns) NVT equilibration simulation with 2.0 fs time step and temperature was set at 300 K. Subsequently, 1ns NPT simulation was performed with 2.0 fs time step to equilibrate the pressure of the system to 1 bar. The LINCS algorithm was used to constrain the bond lengths. The final simulation run of the complexes was performed for 100 ns and trajectories were used to calculate the RMSD, RMSF, Rg, SASA, hydrogen bonds, and PCA. The programming script for each step is provided in (S1, S2 Tables and S1 Fig in S1 File).

## 2.7 Prediction of cross targets

The PharmMapper server was used to identify a wide range of targets using an innovative reverse pharmacophore mapping approach. The best mapping poses of the submitted molecule (.mol2) were aligned against all target proteins present in PharmTargetDB. To perform the reverse pharmacophore matching protocol, the algorithm used comprises a sequential combination of triangle hashing (TriHash) and genetic algorithm (GA) optimization [52]. The candidate targets were ranked based on the calculated highest fit score between the small compound and the pharmacophore models. It was imperative to check the disease-causing potential of the targets.

## 3. Results

### 3.1 Site-directed high throughput virtual screening

The initial docking-based virtual screening result revealed the top 10 hits including 8 compounds from *Bacopa monnieri* and 4 compounds from *Mucuna pruriens* based on the lowest free energy binding score of -8.0 kcal/mol. IMPHY012003 (Betulinic acid), and

**Table 1. Binding affinity estimations of the top 10 compounds and their physiochemical properties.**

| Sl. No. | Compounds | Common Name | Source | Binding Score (kcal/mol) | pKi (nM) | Molecular Weight | Hydrogen bond donor | Hydrogen bond acceptor | Rotational Bond | Ring | Topological Polar Surface Area (TPSA) |
|---|---|---|---|---|---|---|---|---|---|---|---|
| 1. | IMPHY012003 | Betulinic acid | *Bacopa monnieri, Mucuna pruriens* | -9.32 | 146.55 | 456.71 | 2 | 2 | 2 | 5 | 57.53 |
| 2. | IMPHY011880 | Ursolic acid | *Mucuna pruriens* | -9.29 | 156.02 | 456.71 | 2 | 2 | 1 | 5 | 57.53 |
| 3. | IMPHY009537 | Stigmastenol | *Bacopa monnieri* | -9.02 | 246.55 | 414.72 | 1 | 1 | 6 | 4 | 20.23 |
| 4. | IMPHY004141 | alpha-Amyrenyl acetate | *Mucuna pruriens* | -8.92 | 287.71 | 468.77 | 0 | 2 | 2 | 5 | 26.3 |
| 5. | IMPHY007882 | Ebelin lactone | *Bacopa monnieri* | -8.68 | 150.26 | 454.7 | 1 | 3 | 3 | 4 | 46.53 |
| 6. | IMPHY014836 | beta-Sitosterol | *Bacopa monnieri, Mucuna pruriens* | -8.64 | 462.82 | 414.72 | 1 | 1 | 6 | 4 | 20.23 |
| 7. | IMPHY001534 | Jujubogenin | *Bacopa monnieri* | -8.24 | 905.58 | 472.71 | 2 | 4 | 1 | 6 | 58.92 |
| 8. | IMPHY006699 | Bacosine | *Bacopa monnieri* | -8.19 | 993.43 | 456.71 | 2 | 2 | 2 | 5 | 57.53 |
| 9. | IMPHY002708 | Pseudojujubogenin | *Bacopa monnieri* | -8.14 | 1090 | 472.71 | 2 | 4 | 1 | 6 | 58.92 |
| 10. | IMPHY002389 | Bacogenin-A1 | *Bacopa monnieri* | -8.03 | 1300 | 472.71 | 2 | 4 | 2 | 5 | 66.76 |

IMPHY014836 (beta-Sitosterol) were common in top hits from both plants. The ligand efficiency score, docking binding Score, number of conformations in largest clustre, and pKa, were taken into consideration for screening the ligand molecules from the dataset. Docking score of top 10 compounds were fall between -8.03 to -9.32 kcal/mol, which reflect their high degree of binding affinity towards CST. With regard to physiochemical properties, the selected top 10 compounds strictly followed the criteria of Lipinsiki rule of 5 under which molecular mass (<500), number of hydrogen bond donors (<5), hydrogen bond acceptors (<10), and surface area were considered. Positively, all 10 compounds followed the acceptable criteria of rotatable bond of <10, nevertheless, Stigmastenol (IMPHY009537) and beta-Sitosterol (IMPHY014836) with a relatively higher number of rotatable bonds (6), in particular, were able to maintain sufficient flexibility in the active site of the protein. The binding affinity and physiochemical properties of selected compounds are detailed in Table 1.

## 3.2 Absorption, distribution, metabolism, and excretion (ADME) and toxicity analyses

Pharmacokinetic and toxicity analysis of the top 10 hits was performed to ensure the bioavailability and safety of drug-body interaction for future studies [44–46, 53]. Table 2 details the pharmacological profile of the top 10 hits. Intestinal absorption of all 10 compounds was found to be high (>90%) and registered a good caco2 permeability score (>0.90), indicating their high epithelial permeability. Since MLD is a neurodegenerative disease the blood-brain barrier (BBB) permeability was a key consideration. Out of the 10 compounds, only four compounds including Stigmastenol (IMPHY009537), alpha-Amyrenyl acetate (IMPHY004141), Jujubogenin (IMPHY001534) and beta-Sitosterol (IMPHY014836) showed high BBB

permeability with logBB cut-off $\geq$0.3 and high CNS permeability with logPS $\geq$-2, hence they were only considered for further evaluation. Interaction with Renal OCT (organic cation transporter) was used as a parameter to predict the potential deposition and clearance of test compounds from the kidney. On bodily metabolism and excretion criteria, these four compounds stood out positively. These four compounds were found to be non-mutagenic under Ames's test and nontoxic under hepatoxicity. In the ProTOX toxicity class, Stigmastenol, alpha-Amyrenyl acetate, and jujubogenin fall under class '5', while beta-Sitosterol fall under class '4', strongly supporting their 'safe' usage as oral drug.

## 3.3 Interaction analysis

Binding analysis of the top 4 compounds showed their wider occupancy of binding space. With a binding score of -9.02 kcal/mol, Stigmastenol showed the highest affinity towards CST and tightly occupied the binding site by positioning its steroidal backbone on the polar side and interacted with His84, Phe177, Lys85, and Tyr176 via Pi-alkyl and Pi-Pi bond formation in the polar site of the binding pocket. Its branched aliphatic chain with oxygen atom was oriented towards the aromatic region dominated by Ser173, Phe170, and Tyr203 and formed a hydrogen bond with Ser173 while interacting with Phe170, and Tyr203 via pi-alkyl bonds [Fig 1A]. In the CST- alpha-Amyrenyl acetate complex, the five six-membered nonpolar rings flanked by -CH$_3$ groups on each ring interacted with Lys85, His84, His141, and Phe177 via Pi-alkyl interaction while the polar aliphatic fragment interacted with the Lys82, Ser88, and Arg282 [Fig 1B]. In the CST-beta-Sitosterol complex, the polar steroidal backbone of the compound positioned in the polar region and interacting with His84, Lys85, and Tyr176 via Pi-alkyl and hydrogen bond formation, while the nonpolar aliphatic branched moiety of the compound positioned towards the aromatic side and interacted with Tyr203, and Phe170 via Pi-alkyl bonds [Fig 1C]. In the CST-Jujubogenin complex, Jujubogenin with five six-membered aromatic rings with an oxygen atom at one end interacted with Asn113 by forming a hydrogen bond at the polar site, while at the middle of the active site pocket the compound flanked by His84 and His141 via forming Pi-sigma and Pi-alkyl interactions, while the aliphatic chain on the other side of the compound interacted with Phe170 and Tyr203 and forming a hydrogen bond with Ser173 [Fig 1D]. Table 3 provides in-depth details of non-bonded interaction of each protein-ligand complex.

**Table 2. Pharmacological profile of the top 10 molecules using pKCSM, SwissADME, and ProTox webserver.**

| Sl. No. | Compound ID | Absorption | | Distribution | Metabolism | | Excretion | | Toxicity | | | |
|---|---|---|---|---|---|---|---|---|---|---|---|---|
| | | Human Intestinal Absorption | Caco2 permeability | BBB permeation | Cytochrome P450 2D6 inhibitor | Cytochrome P450 2D6 substrate | Renal OCT2 substrate | AMES | Hepato-toxicity | Predicted LD50 | Predicted Toxicity class |
| 1. | IMPHY012003 | High | High | Moderate | No | No | No | No | Yes | 2610mg/kg | 5 |
| 2. | IMPHY011880 | High | High | Moderate | No | No | No | No | Yes | 2000mg/kg | 5 |
| 3. | IMPHY009537 | High | High | High | No | No | No | No | No | 5000mg/kg | 5 |
| 4. | IMPHY004141 | High | High | High | No | No | No | No | No | 3460 mg/kg | 5 |
| 5. | IMPHY007882 | High | High | Moderate | No | No | No | No | Yes | 5000mg/kg | 5 |
| 6. | IMPHY014836 | High | High | High | No | No | No | No | No | 890mg/kg | 4 |
| 7. | IMPHY001534 | High | High | High | No | No | No | No | No | 5000mg/kg | 5 |
| 8. | IMPHY006699 | High | High | Poor | No | No | No | No | Yes | 2500mg/kg | 5 |
| 9. | IMPHY002708 | High | High | Poor | No | No | No | No | No | 8000mg/kg | 6 |
| 10. | IMPHY002389 | High | High | Moderate | No | No | No | No | No | 238mg/kg | 3 |

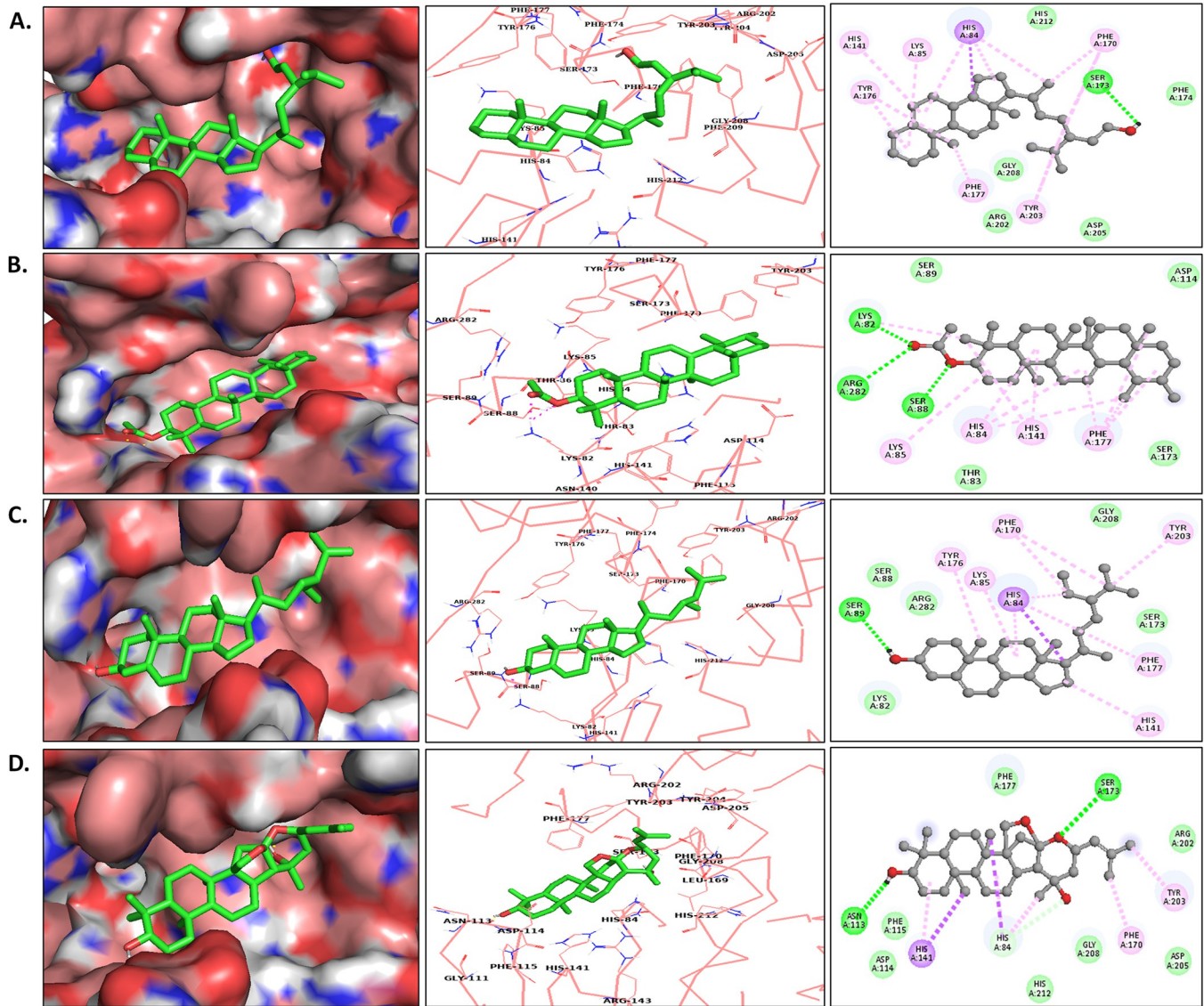

**Fig 1.** Non-bonded interaction of top four docked complexes, including surface view, 3D views, and 2D view of protein-ligand interaction at the substrate binding site in (A) CST- Stigmastenol, (B) CST-alpha-Amyrenyl acetate (C) CST-beta-sitosterol and (D) CST-Jujubogenin complexes.

## 3.4 MM-PB(GB)SA analysis of top 4 hits for binding energy calculation

Further, MM/PB(GB)SA analysis was performed to evaluate binding energy of protein-ligand interaction in terms of molecular mechanics using Fast Amber Rescoring server. In MM-PB (GB)SA study, the binding score was found to be in the order of Beta-Sitosterol > Stigmastenol > Alpha-Amyrenyl acetate > Jujubogenin, which signify the binding affinity towards CST. Overall, the better binding score obtained by MM-PB(GB)SA analysis than by molecular docking reaffirmed the rigidity and stability of protein-ligand complexes. Despite the deviation from the docking scores for each complex, the MM/GBSA results supported the candidature of top hits for potential drug against CST. Table 4 provides the details based on different binding free energy parameters.

**Table 3. Pre-MD non-bonded interaction between CST and top four hits.**

| IMPPAT ID | Residue in contact | Interaction types | Distance in Å |
|---|---|---|---|
| Stigmastenol | Ser173 | Conventional Hydrogen Bonds | 2.10 |
| | His84 | Pi-Sigma | 4.20 |
| | His84 | Pi-Alkyl | 4.87 |
| | His84 | Pi-Alkyl | 5.08 |
| | Phe170 | Pi-Alkyl | 4.63 |
| | Tyr176 | Pi-Alkyl | 5.07 |
| | Tyr176 | Pi-Alkyl | 5.48 |
| | Phe177 | Pi-Alkyl | 5.15 |
| | Tyr203 | Pi-Alkyl | 3.09 |
| Alpha-Amyrenyl acetate | Lys82 | Conventional Hydrogen Bonds | 2.20 |
| | Ser88 | Conventional Hydrogen Bonds | 2.10 |
| | Arg282 | Conventional Hydrogen Bonds | 3.40 |
| | His84 | Pi-Alkyl | 4.98 |
| | Lys85 | Pi-Alkyl | 6.76 |
| | His141 | Pi-Alkyl | 4.0 |
| | Phe177 | Pi-Alkyl | 6.21 |
| Beta-Sitosterol | Lys82 | Conventional Hydrogen Bonds | 2.00 |
| | Ser89 | Conventional Hydrogen Bonds | 3.06 |
| | His84 | Pi-Sigma | 4.26 |
| | His84 | Pi-Alkyl | 4.37 |
| | His84 | Pi-Alkyl | 5.33 |
| | Lys85 | Pi-Alkyl | 6.01 |
| | Phe170 | Pi-Alkyl | 5.01 |
| | Phe170 | Pi-Alkyl | 5.67 |
| | Tyr176 | Pi-Alkyl | 6.09 |
| | Phe177 | Pi-Alkyl | 5.90 |
| | Tyr203 | Pi-Alkyl | 4.16 |
| Jujubogenin | Asn113 | Conventional Hydrogen Bonds | 1.90 |
| | Ser173 | Conventional Hydrogen Bonds | 2.20 |
| | His84 | Pi-Sigma | 3.63 |
| | His84 | Van der Waal | 5.41 |
| | His84 | Pi-Alkyl | 6.05 |
| | His141 | Pi-Sigma | 4.34 |
| | His141 | Pi-Alkyl | 6.41 |
| | Phe170 | Pi-Alkyl | 5.76 |
| | Tyy203 | Pi-Alkyl | 4.01 |

**Table 4. Calculated binding energy for each of the CST complexes with selected compounds via MM-PB(GB)SA approach using Fast Amber Rescoring server.**

| Compounds | AutoDock Binding affinity (kcal/mol) | MM-PBSA binding free energy (kcal/mol) | | | | | |
|---|---|---|---|---|---|---|---|
| | | PB3 | PB4 | GB1 | GB2 | GB5 | GB6 |
| Stigmastenol | -9.02 | -18.5 | -20.22 | -30.33 | -26.59 | -30.54 | -28.37 |
| Alpha-Amyrenyl acetate | -8.92 | -8.04 | -13.15 | -33.77 | -26.98 | -28.63 | -16.62 |
| Beta-Sitosterol | -8.64 | -21.86 | -24.01 | -40.29 | -33.88 | -37.48 | -28.14 |
| Jujubogenin | -8.24 | -19.8 | -19.93 | -28.12 | -25.96 | -29.37 | - |

### 3.5 Molecular dynamic simulation

The molecular dynamic simulation uncovered the dynamic behavior of protein-ligand complexes and determined their degree of stability under aqueous environment [54–56]. Each protein-ligand complex was simulated for 100 ns time scale and trajectory was analyzed comparing with free CST protein and CST-substrate (GC) complex.

**3.5.1 Structural deviations and compactness.** Root mean square deviation (RMSD) measures the conformational changes of the protein under different complex formations from its initial position to its final position during the simulation timespan [57–59]. The RMSD graph of the CST- Beta-Sitosterol complex achieved the fastest initial stability after 5 ns while complexes of Stigmastenol and Jujubogenin showed stability after 20 ns. The CST- Alpha-Amyrenyl acetate complex showed overall stability with minor fluctuation at 50 ns. The average deviation was found for CST, CST-GC, CST-Jujubogenin, CST- Alpha-Amyrenyl acetate, CST- Stigmastenol, and CST-Beta-Sitosterol were 0.49, 0.76, 0.63, 0.54, 0.64, and 0.64, respectively. Under substrate binding, a slight structural deviation was observed in the CST structure. Positively, the CST complex with all selected compounds confined their structural deviation within the limit of RMSD of the free CST and the CST-substrate complex RMSDs [Fig 2I(A)-

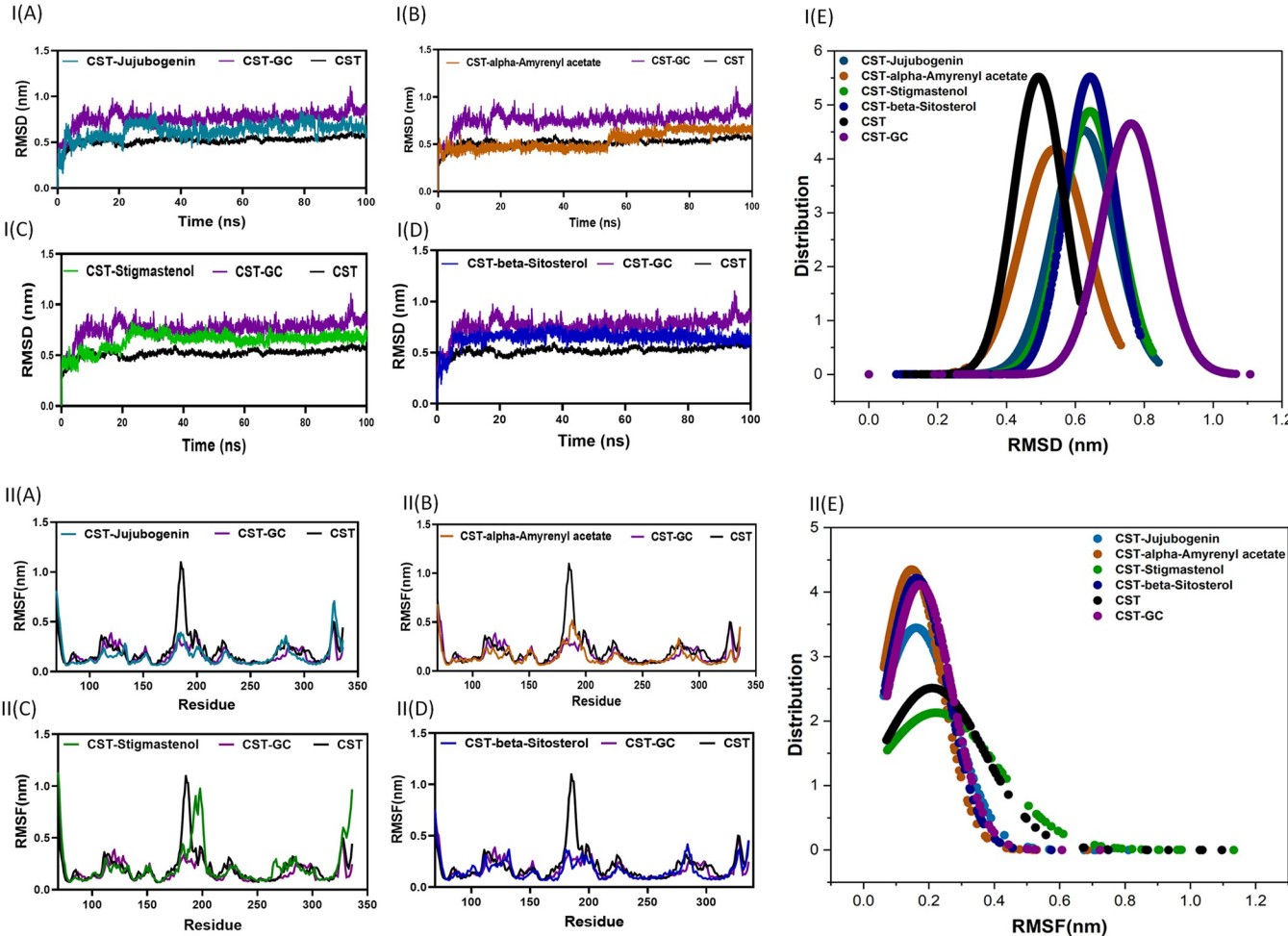

**Fig 2. The representation of structural deviations of CST involves the analysis of the root mean square deviation (RMSD) plot over time and the root mean square fluctuation (RMSF) of residues in CST and its complexes.** The RMSD plot, shown in panel (I), displays the changes in the spatial structure of CST over a specific time scale, highlighting any deviations from its original conformation. In panel (II), the RMSF distribution displays the magnitude of fluctuations of each residue in the protein structure. I(E) and II(E) represent the probability distribution functions of RMSD and RMSF plots.

2I(D)]. Probability distribution function [Fig 2I(E)] of RMSD values showed the overall stability of complexes without any major shift.

The root mean square fluctuation (RMSF) measured the average divergence of particles from the reference position, showing regions with higher fluctuation and the impact of ligand binding on those regions throughout the simulation time [60, 61]. The residue-based fluctuation was observed in free CST in the region of 175–190 amino acid residues. Among four hits, similar fluctuation in the region between 180–200 was observed significantly in CST complexed with Stigmastenol as shown in Fig 2II(C). The observation of the RMSF of the other three hits was closer to the RMSF of the CST-substrate complex. The average RMSF of free CST, CST-substrate complex, CST-Jujubogenin, CST-alpha-Amyrenyl acetate, CST-Stigmastenol, and beta-Sitosterol were 0.15, 0.17, 0.16, 0.15, 0.19, and 0.16, respectively. Fig 2II(A)-2II (D) represents the RMSF graph of the residual fluctuation of CST protein complexed with selected compounds. Fig 2II(E) represents the probability distribution function of RMSF which shows that CST-alpha-Amyrenyl acetate and CST-Stigmastenol had closer distribution to CST-GC complex and thus indicates their capacity to competitively inhibit CST.

The radius of gyration (Rg) measures the structural compactness and overall conformational stability with ligand binding [62–64]. The average radius of gyration of the free CST was 1.76 nm which was reduced to 1.71 nm after binding to the substrate. This slight reduction of Rg in the protein-substrate complex is attributed to increasing the compactness of the CST structure by creating room for properly adjusting the substrate to the substrate binding site by pushing the residues inward. A similar trend was observed in the CST complexed with selected compounds (Fig 3(A), upper panel). The average Rg of CST-Jujubogenin, CST-alpha-Amyrenyl acetate, CST- Stigmastenol, and beta-Sitosterol were 1.73 nm, 1.68 nm, 1.74 nm, and 1.73, respectively. The lowest Rg of CST-alpha-Amyrenyl acetate complex suggests that the compactness of CST protein was relatively higher, while the compactness of all other selected compounds was found between the free protein and the protein-substrate complex based on Rg data. The probability distribution function of Rg indicated not a major shift in the structural compactness of CST under different complex formations when compared to the protein-substrate complex (Fig 3(A), lower panel).

Further, the solvent-accessible surface area (SASA) analysis measured the impact of ligand binding on the protein exposed area to the surrounding solvent [65–67]. The average SASA of free CST was 180.1 nm$^2$ which was reduced to 173.07 nm$^2$ when complexed with the substrate due to compactness of the protein. A similar trend was observed in CST complexed with selected compounds. The SASA was found in the order of beta-Sitosterol (174.73 nm$^2$) > CST-Jujubogenin (173.27 nm$^2$) > CST-Stigmastenol (172.08 nm$^2$) > CST-alpha-Amyrenyl acetate (171.49 nm$^2$), and, which signify that Stigmastenol and alpha-Amyrenyl acetate should have better structural compactness as they have relatively lesser solvent accessibility. Other two protein-ligand complexes also maintained their solvent accessibility area closer to CST-substrate complex (Fig 3(B), upper panel). Further, in the probability distribution graph of SASA suggested the stability of complexes were maintained with minor increase in CST- alpha-Amyrenyl acetate complex (Fig 3(B), lower panel).

**3.5.2 Hydrogen bonds dynamics.** Hydrogen bond analysis was imperative to ensure the structural stability of CST protein with selected ligands [68, 69]. A dynamic analysis of free CST, CST-substrate complex and CST complex with selected ligands was conducted to check the consistency of intramolecular hydrogen bonds of CST. The average intramolecular hydrogen bonds for free CST, CST-GC, CST-Jujubogenin, CST-alpha-Amyrenyl acetate, CST- Stigmastenol, and beta-Sitosterol were 175.37, 186.13, 172.20, 180.62, 174.09, and 175.8, respectively. This suggests that there were no significant changes in the number of hydrogen bonds within the CST protein in complex with Jujubogenin, Stigmastenol, and beta-Sitosterol,

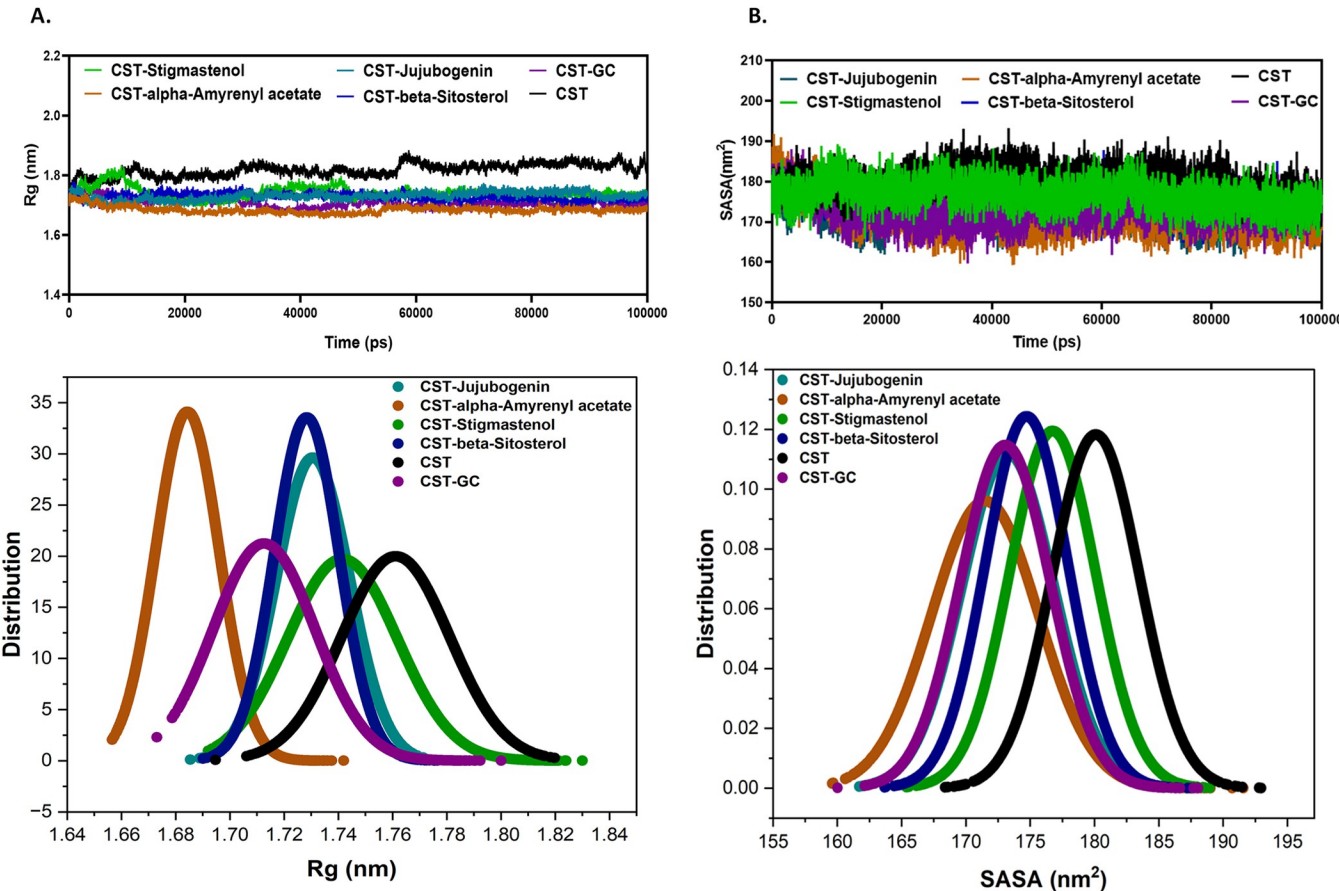

**Fig 3.** The structural compactness of CST using (A) Rg and (B) SASA plots. The lower panels represent the Probability Density Function of the values to understand the average distribution pattern the Rg and SASA values of CST and its complexes.

while intramolecular hydrogen bonds slightly increased in the CST-alpha-Amyrenyl acetate complex, indicating a relatively more compactness of protein structure in presence of ligand (Fig 4(I)–4(IV), upper panel). This increased structural compactness correlates with the result of Rg and SASA. The probability distribution function of intramolecular hydrogen bonds also showed the stability of the protein-ligand system (Fig 4(I)–4(IV), lower panel).

Furthermore, the intermolecular atomic level interaction was analyzed to predict the binding strength of selected compounds in the substrate binding pocket throughout the simulation [70]. The range of hydrogen bond between the selected compounds and CST active site residues was between 1–4. Jujubogenin and beta-Sitosterol continuously interacted with protein via multiple hydrogen bonds throughout the simulation, suggesting their higher rigidity, while CST-Stigmastenol could maintain one hydrogen bond number throughout the simulation [Fig 4B(I), 4B(III), 4B(IV)]. CST-alpha-Amyrenyl acetate showed discontinued intermolecular hydrogen bonds throughout the simulation [Fig 4B(II)], however its continuous positioning in the binding pocket can be attributed to other noncovalent interactions.

**3.5.3 Principal component analysis.** Principal component analysis (PCA) was performed to analyze key conformational changes during ligand binding [71–74]. The overall motion of the CST protein under different complex formation was determined primarily by first two and first three eigenvectors that reflected the overall dynamics of the molecular subspace of the protein in the presence of ligand binding [Fig 5(A) and 5(B)]. The motion of CST-substrate

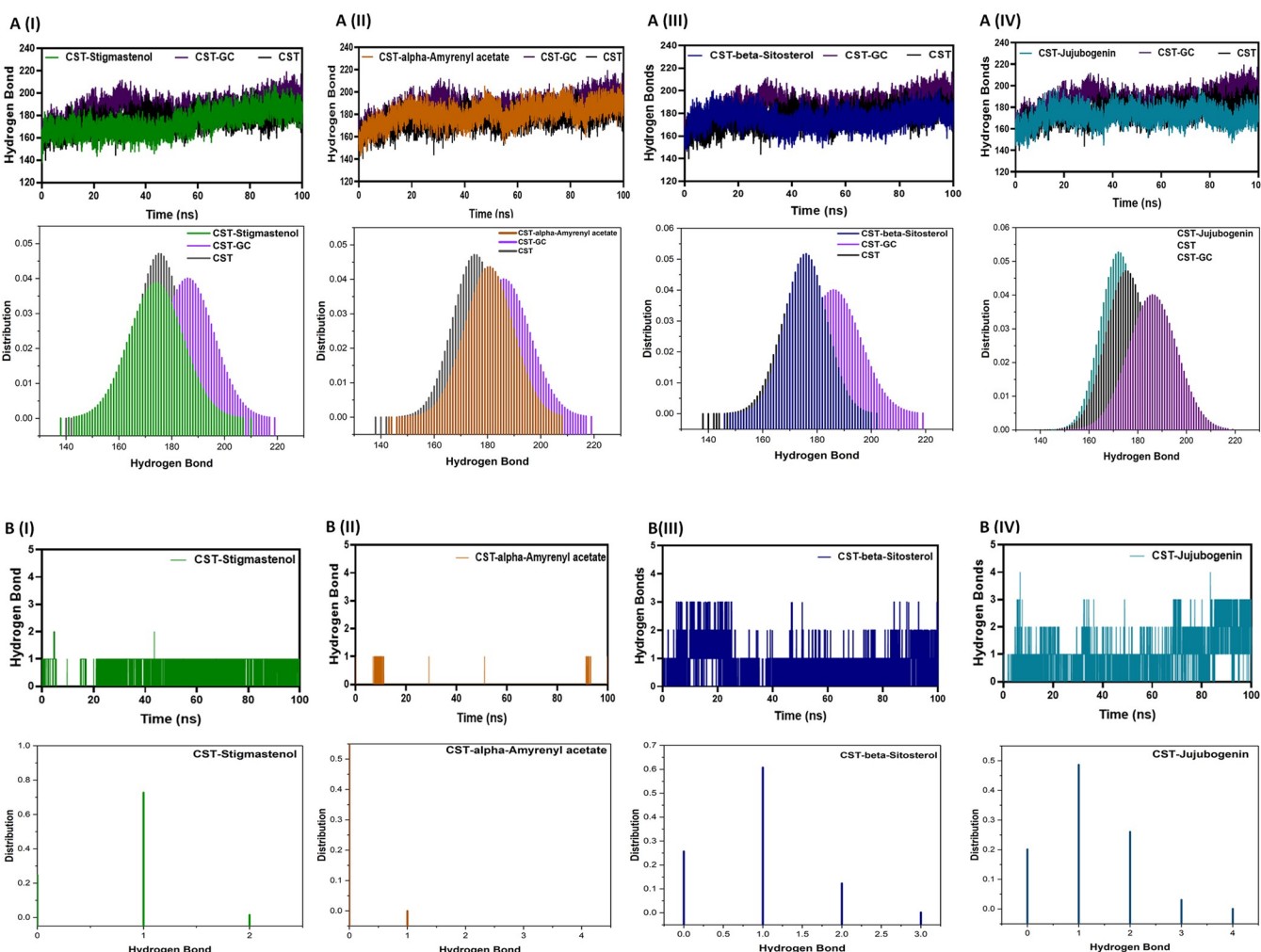

**Fig 4.** **(A)** The hydrogen bond dynamics of CST in complex with selected compounds. (A) intramolecular H-bonds **(B)** Intermolecular hydrogen bonds of protein-ligand complex. I, II, III, and IV represent the hydrogen bonds between protein and ligand in a protein-ligand complex of Stigmastenol, alpha-Amyrenyl acetate, beta-Sitosterol, and Jujubogenin, respectively. The lower panel of I, II, III, IV represents the probability density function of respective complexes.

complex was little dispersed as compared to free CST. The PCA of each complex had different motion pattern. Among the four hits, alpha-Amyrenyl acetate and Stigmastenol showed cluster and compact type of motion, covering the range from -5 to 2.6 nm for eigenvector 1, from -4 to 3 for eigenvector 2 and from -2 to 2 nm for eigenvector 3 in CST-Stigmastenol complex, and range of -3 to 3.4 nm for eigenvector 1, from -2 to 2.3 for eigenvector 2, from -1.5 to 2 nm for eigenvector 3 in CST-alpha-Amyrenyl acetate complex. With eigenvector 1 ranged between -2 and 4, eigenvector 2 ranged between -1.5 and 5, and eigenvector 3 from -2.5 to 1.5, the CST-beta-Sitosterol complex also showed good compact motion behavior. With eigenvector 1 valued from −8 to 2 nm, eigenvector 2 from -4 to 5, and eigenvector 3 from −3.0 to 2.0 nm, the CST- Jujubogenin complex was found to be the most dispersed among four. Nevertheless, all selected compounds maintained the minimal essential conformational subspace and their eigenvector were within the range of free CST and CST-GC complex.

**3.5.4 Free energy landscape analysis.** The free energy landscape (FEL) was calculated using the first two principal components to monitor the distinct binding conformation while

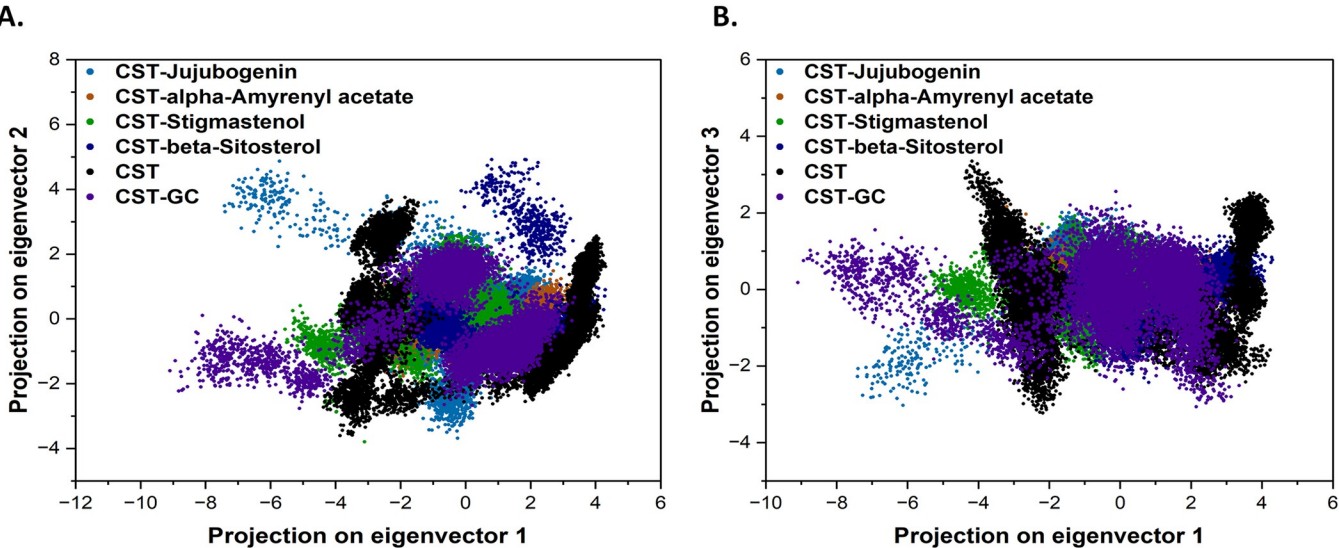

**Fig 5.** Two-dimensional projections of the protein-ligand conformational changes throughout the simulation trajectory using (A) the first two eigenvectors, (B) the first three eigenvectors. Black, violet, light blue, brown, green, and dark blue color denotes free CST, CST-GC, CST-Jujubogenin, CST-alpha-Amyrenyl acetate, CST-Stigmastenol, and CST-beta-sitosterol complexes, respectively.

highlighting the most dominant internal mode of motion [75–78]. The dark blue represented the most energetically favored region while yellow represented the relatively unfavorable region and high energy state. With a global energy minimum of 16.60 kJ/mol, the free CST represented the stable FEL with relatively increased conformational space with requirement of the lowest free energy. Substrate binding slightly increased the global energy minimum to 17.10 kJ/mol. As per the FEL graph, CST-alpha-Amyrenyl acetate and CST-Stigmastenol were found to be relatively better stable complexes with a wider blue zone with global minima of 14.60 kJ/mol and 14.40 kJ/mol, respectively, which were relatively lower than global minima of Jujubogenin (17.50 kJ/mol) and beta-Sitosterol (17.80 kJ/mol) (Fig 6).

### 3.6 Interaction analysis of best compounds from *Bacopa monnieri* and *Mucuna pruriens*

PostMD interaction analysis revealed the firm affinity of the selected compounds to maintain their hold in the protein binding pocket of the protein under a continuous dynamic aqueous environment. As earlier, Stigmastenol maintained its position well in the binding site by keeping its steroidal backbone horizontally at the left polar site of the binding pocket by interacting with His84, His141, Phe177, and Tyr176 while its aliphatic chain interacts with Phe170 and Tyr203 via Pi-alkyl interactions in the right side of pocket (Fig 7(A)). alpha-Amyrenyl acetate maintained its position at the binding site with a little shift of the hydrogen bond interaction with Lys85 from Lys82 of pre-MD interaction. This shift was because of the orientation of the alkyl branch at the 'oxygen' atom due to the presence of rotatable bonds. The steroidal backbone was surrounded by His84 and His141 from both sides and interacted via Pi-sigma, Pi-Pi, and Pi-alkyl bond formations. The rightmost ring of the compound interacted with Phe170 and Ala211 via Pi-alkyl bonds (Fig 7(B)). In the CST-beta-Sitosterol complex, the compound retained its positioning in the binding site pocket as in the docked complex, however, its side chain showed little shift in its positioning (Fig 7(C)). In the CST-Jujubogenin complex, the compound occupied the binding pocket in a similar orientation as in the docked conformation and interacted with HIS84 at the middle via hydrogen bond and Pi-sigma bonds. At the polar

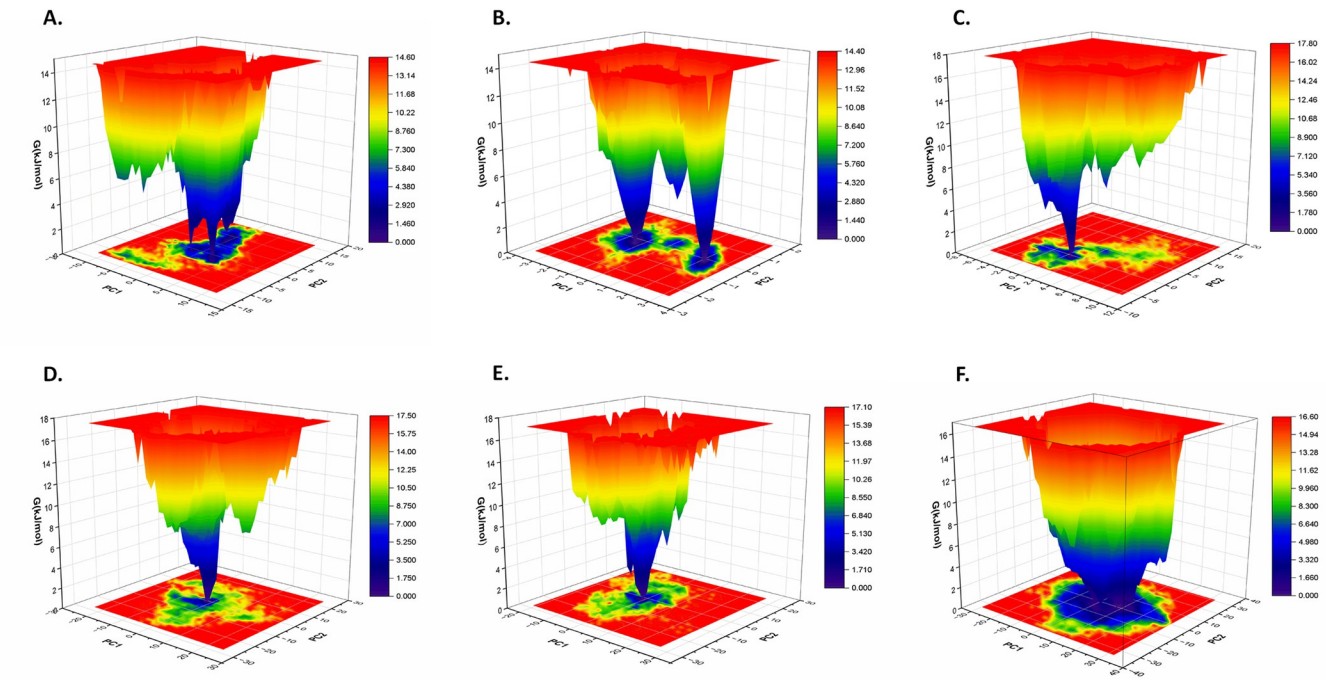

**Fig 6.** Free energy landscape created from 100 ns MD simulation trajectories of (A) CST- CST- Stigmastenol, (B) CST-alpha-Amyrenyl acetate, (C) CST- beta-Sitosterol, and (D) CST-Jujubogenin (E) CST-GC and, (F) free CST. The color bar denotes the relative free energy value from dark blue (lowest) to dark yellow (highest).

site, the compound interacted with Phe177 through Pi-alkyl bonding while at the aromatic site Tyr203, and Phe170 were involved in the interaction via Pi-alkyl bond formation (Fig 7(D)). Table 5 provides details of non-bonded post-MD interaction analysis of each complex.

## 3.7 Cross target identification and reverse pharmacophore mapping of 4 top hits using PharmMapper

Finally, the selected compounds were subjected to cross-target identification as a part of the reverse pharmacophore mapping approach using PharmMapper with fitness score cutoff of 5.0. The target selection was performed keeping the nervous system disorders in focus. Among four compounds, Stigmastenol showed the least cross-contamination with other targets. The only potential target of Stigmastenol was found to be nuclear receptor ROR-alpha, which has not been shown to be indicated by any disease. Transthyretin and Retinol-binding protein 4 were key common targets among alpha-Amyrenyl acetate, Jujubogenin and beta-Sitosterol. Table 6 provides the details of the cross-target identification and reverse pharmacophore mapping of 4 top hits.

## 4. Discussion

This study describes the identification and characterization of potent and selective drugs against CST to counter the accumulation of sulfatides responsible for myelin sheath disturbances in the nervous system as part of the strategy to develop substrate reduction therapy for MLD. Towards the development of SRT, a major breakthrough was the development of a three-dimensional computational model of the CST protein by our group [12]. The present study used this 3D model to screen phytochemicals of *Bacopa monnieri* and *Mucuna pruriens*

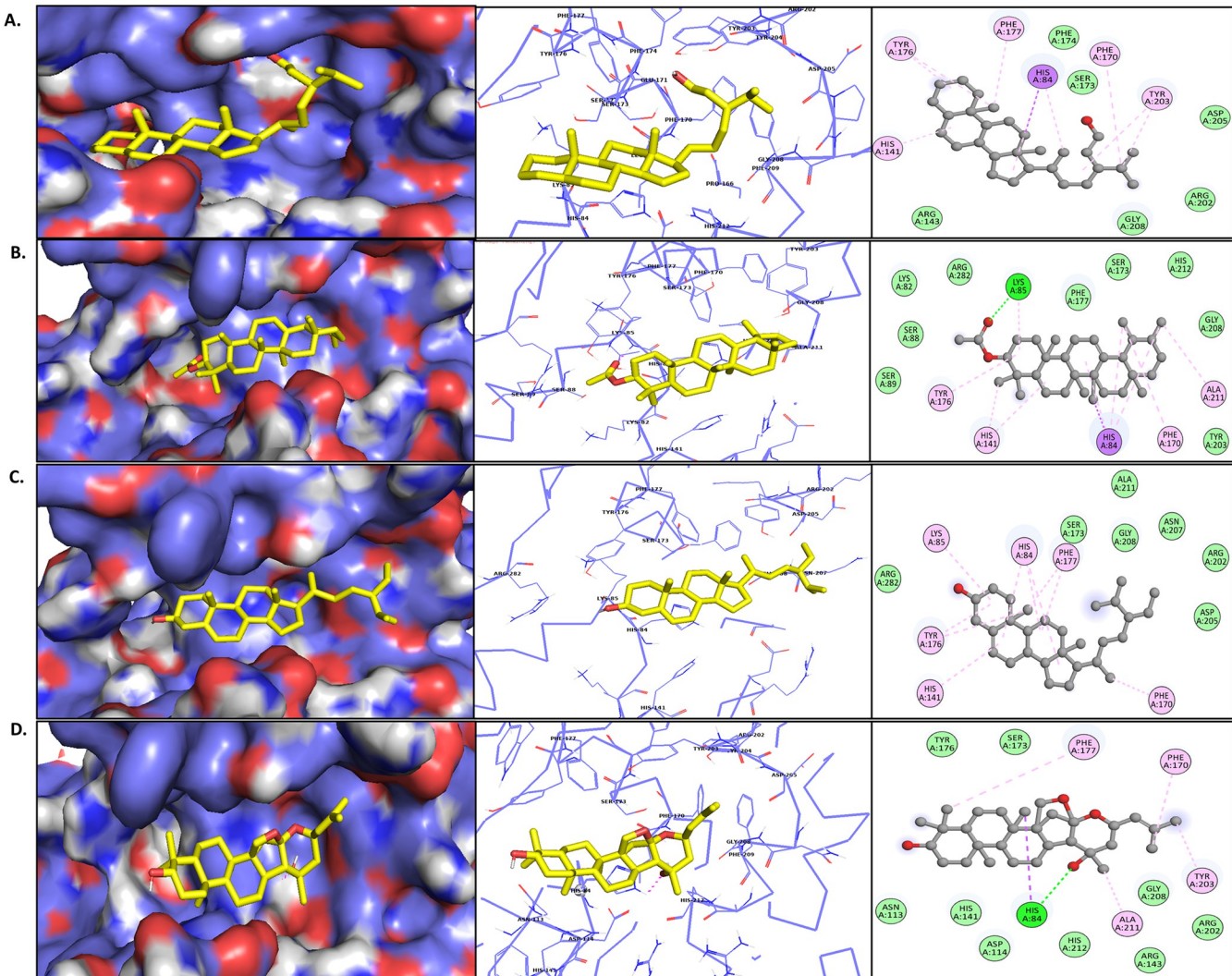

**Fig 7.** Post MD interaction Analysis of selected compounds in substrate site of CST with a surface view, 3D protein-ligand interaction, and 2D representation of the interaction of (A) CST-Stigmastenol complex, (B) CST- alpha-Amyrenyl acetate complex, (C) CST-beta-Sitosterol complex, and (D) CST-Jujubogenin. Conventional hydrogen bonds, Pi-sigma bonds, Pi-alkyl bonds, and Vander Waal interactions are represented in dark green, violet, light pink, and light green colors respectively.

for selective and potent drug candidates against CST. Based on the binding score cutoff -8.0 kcal/mol, 10 compounds were considered, and then, in order to reduce the false positive results, some specific pharmacokinetic parameters were applied to screen the best drug candidates for the targeted brain disorder. Four top hits (Stigmastenol, alpha-Amyrenyl acetate, beta-Sitosterol, and Jujubogenin) qualified the blood-brain barrier permeability criteria which was considered critical to increase the availability of drugs to neurons for regulating catalytic action of CST. Overall, with no significant toxicity, these four compounds were found to have good absorption, distribution, excretion, and metabolism, thus, they are assumed to be safe for future studies. Furthermore, the MM-PB(GB)SA scoring study confirmed the efficiency of selected compounds towards CST and the rigidity of protein-ligand complexes, which was prerequisite for inhibition of the target protein.

**Table 5. Post MD simulation non-bonded interaction analysis of protein-ligand complex with top four hits.**

| IMPPAT ID | Residue in contact | Interaction type | Distance in Å |
|---|---|---|---|
| Stigmastenol | His84 | Pi-Sigma | 4.65 |
| | His84 | Pi-Alkyl | 5.77 |
| | His84 | Pi-Alkyl | 5.96 |
| | His141 | Pi-Alkyl | 6.96 |
| | Phe170 | Pi-Alkyl | 6.69 |
| | Tyr176 | Pi-Alkyl | 5.68 |
| | Tyr176 | Pi-Alkyl | 5.78 |
| | Phe177 | Pi-Alkyl | 5.73 |
| | Tyr203 | Pi-Alkyl | 4.36 |
| alpha-Amyrenyl acetate | Lys85 | Conventional Hydrogen Bonds | 2.00 |
| | His84 | Pi-Sigma | 4.24 |
| | His84 | Pi-Alkyl | 5.35 |
| | His84 | Pi-Alkyl | 5.73 |
| | His141 | Pi-Alkyl | 4.27 |
| | Phe170 | Pi-Alkyl | 5.22 |
| beta-Sitosterol | His84 | Pi-Alkyl | 5.24 |
| | His84 | Pi-Alkyl | 5.37 |
| | His84 | Pi-Alkyl | 6.35 |
| | Phe170 | Pi-Alkyl | 6.14 |
| | Tyr176 | Pi-Alkyl | 4.70 |
| | Tyr176 | Pi-Alkyl | 5.43 |
| | Phe177 | Pi-Alkyl | 5.57 |
| Jujubogenin | His84 | Conventional Hydrogen Bonds | 2.80 |
| | His84 | Pi-Sigma | 3.97 |
| | Phe170 | Pi-Alkyl | 6.45 |
| | Phe177 | Pi-Alkyl | 6.38 |
| | Tyy203 | Pi-Alkyl | 4.99 |

**Table 6. Reverse pharmacophore mapping for identifying potential targets of top 4 hits among nervous system proteins.**

| Compounds | Proteins | PDB ID | Disease | No. of Pharma-cophore features | Fitness Score |
|---|---|---|---|---|---|
| Jujubogenin | Estradiol 17-beta-dehydrogenase 1 | 1JTV | None | 8 | 5.52 |
| | Nuclear receptor ROR-alpha | 1S0X | None | 10 | 5.227 |
| | Hepatocyte growth factor receptor | 3F82 | None | 11 | 5.147 |
| | Retinol-binding protein 4 | 1RBP | Night vision problems | 8 | 5.114 |
| | Aldo-keto reductase family 1 member C3 | 1RY0 | None | 7 | 5.074 |
| | Transthyretin | 1RLB | Amyloidosis type 1 (AMYL1) | 10 | 5.003 |
| alpha-Amyrenyl acetate | Transthyretin | 1RLB | Amyloidosis type 1 (AMYL1) | 10 | 5.722 |
| | Alpha-tocopherol transfer protein | 1R5L | None | 12 | 5.472 |
| | Retinol-binding protein 4 | 1RBP | Night vision problems | 8 | 5.29 |
| Stigmastenol | Nuclear receptor ROR-alpha | 1S0X | None | 10 | 6.549 |
| beta-Sitosterol | Nuclear receptor ROR-alpha | 1S0X | None | 10 | 5.875 |
| | Transthyretin | 1RLB | Amyloidosis type 1 (AMYL1) | 10 | 5.573 |
| | Cellular retinoic acid-binding protein 2 | 1CBS | None | 10 | 5.533 |
| | Retinol-binding protein 4 | 1RBP | Night vision problems | 8 | 5.046 |

The detailed interaction analysis showed a strong binding affinity of the compounds towards the active site of the protein. These compounds were properly placed by interacting with two subsites of the active site, the polar site on the left and the aromatic site on the right, mediated by two histidines- His84 and His141 which were positioned parallel to each other and surround the compound from both sides in the middle of the substrate binding site. Lys85, Tyr176, and Phe177 were key residues in the polar site that were crucial for placing the steroidal backbones in the proper orientation. Ser173, Phe170, and Tyr203 were crucial residues in the aromatic site of the active site. In contrast to the other three complexes where the compound's steroidal backbone was positioned vertically, in the CST-Stigmastenol complex, the steroidal backbone of the compound was positioned horizontally flat on the left side of the active site and interacted by Pi-Pi and Pi-alkyl bonds with neighboring residues. This horizontal positioning of the steroidal backbone gave conformational stability to the protein-ligand complex throughout the simulation. While in other three, the probable reason for vertical orientation of the steroidal backbones was the presence of oxygen atom which facilitated interaction with polar residues which can be evident by hydrogen bond interaction with Lys82, Ser89, and Arg282 in CST- alpha-Amyrenyl acetate complex, Ser89 and Lys82 in CST- beta-Sitosterol complex, and Asn113 in CST-Jujubogenin complex. Apart from hydrogen bonding, Pi-Pi, Pi-sigma, Pi-alkyl, and Van der Waal interaction helped to hold the molecule in that site. Although the four selected compounds maintained their grip in the binding pocket throughout the simulation, a little upward shift of beta-Sitosterol was observed in the active site due to the presence of long aliphatic branched fragments with multiple rotatable bonds that caused less rigidity to the complex Fig 8(A)–8(D)). The post-MD docked complex revealed that Stigmastenol and alpha-Amyrenyl acetate were better aligned and occupied the binding pocket as the docked conformations (Fig 8(A) and 8(B)). Without drastic changes, these conformational adjustments showed the behavior of compounds in the active site under dynamic environmental conditions while maintaining the structural integrity of the protein.

The trajectory analysis of simulated protein-ligand complexes further endorsed the stability and integrity of compounds in the binding pocket. RMSD of these protein-ligand complexes did not exceed 1.0 nm and was between the RMSD range of the free CST and the CST-substrate complex. RMSD and RMSF studies confirmed the overall stability of the four hits. Rg and SASA analysis confirmed the structural compactness of protein-ligand complexes closer to the CST-substrate complex with no major shift observed in the structural pattern after

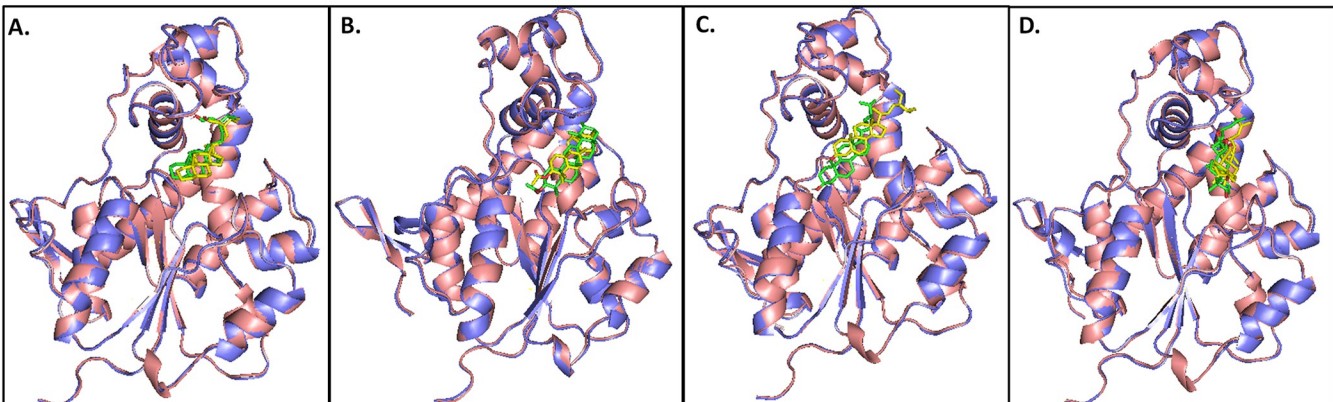

**Fig 8.** The superimposition between pre- and post-MD complexes of Stigmastenol, alpha-Amyrenyl acetate, beta-Sitosterol, and Jujubogenin complexes (A-D). The light pink color indicates the pre-molecular dynamics structure with green colored ligands, and the blue color indicates the post-molecular dynamics structure with yellow ligands.

ligand binding. Overall, this was a good sign for the development of competitive inhibitors for CST catalysis. The hydrogen bonds analysis was also supported by the result of other descriptors from MD simulation with minimal over-fluctuation across the simulation time and endorsed the overall compactness of the protein-ligand complexes. Thus, all four hits showed stable complex formation with relatively better performance observed in the case of CST-Stigmastenol and CST-alpha-Amyrenyl acetate complexes, occupying the minimal conformational space by principal components while maintaining their global minima relatively better than that of the free CST. Thus, in-depth trajectory analysis provides a wholesome idea about the behavior of compounds inside the active site component, despite that the chances of false negatives at the experimental level cannot be avoided.

Furthermore, reverse target analysis revealed that the compound Stigmastenol could be the safest drug candidate as it did not interact with any other potential target. Transthyretin and Retinol binding protein 4 (RBP4) was found to be common potential targets for alpha-Amyrenyl acetate, beta-Sitosterol, and Jujubogenin. In the nervous system, Transthyretin facilitates the transport of thyroxin for myeline sheath formation [79, 80]. Cross-binding with Transthyretin can lead to transthyretin amyloidosis (ATTR), a progressive disorder disturbing the nervous system by accumulation of amyloid-β (Aβ) [81]. RBP4 can also bind to transthyretin and reduce its amyloid-β lowering capacity. While the cross interaction of compounds with RBP4 may have the capacity to reduce amyloidosis by restricting RBP4 to interact with Transthyretin [82]. The cross-interaction of compounds with RBP4 could be beneficial to the nervous system. Therefore, the cross-interaction with Transthyretin and RBP4 needs to be checked via *in vitro* and *in vivo* studies. In summary, the combinatorial high throughput computational studies showed the overall stability of selected compounds in the binding site pocket of *cerebroside sulfotransferase*. To understand the merit of existing discrepancies found in reverse pharmacophore mapping, these compounds need further experimental validation through *in vitro* and *in vivo* studies. Since the CST sequence and CST full length cDNA clones are available that can be exploited for cloning, expression and purification of CST protein for *in vitro* enzyme based inhibition study and cell-line based screening. This would be a directional approach to reach to the lead molecule.

Thus, in line with recent developments in bioinformatics, computational screening has become essential to reduce the failure of drug candidates during the preclinical phase of drug discovery and thereby save time and cost and more importantly, give a right direction to the development of potent drugs. However, *in silico* approaches in MLD was not immune to challenges. Although the challenge of CST structure unavailability was effectively addressed with the development of a 3D homology model of CST [12], in-depth structural validation through crystallography remains an important area on which future research can focus. Another big challenge in developing SRT is finding suitable ligand scaffold for protein active site. *In silico*-based screening of phytoconstituents of widely known neuroprotective medicinal plants including *Bacopa monnieri* and *Mucuna pruriens*, may have potential to provide selective compounds for CST inhibition, as these plants are enriched with unique set of phytochemicals which have been proactive towards cognitive functioning. Molecular docking, ADME and in-depth MD based trajectory analysis provide sound insight about the behavior of compounds inside the active site pocket with reducing the possibilities of inherent challenges of different stages of high throughput screening including data availability, consistency, accuracy of atomistic model, improper trajectory analysis, etc. [83, 84]. Despite applying the multilevel screening approach, the chances of false negative at experimental level cannot be avoided. Therefore, further *in vitro* and subsequent *in vivo* validation become imperative for ensuring the feasibility of selected compounds at preclinical level. The present study thus initiates the gamut of preclinical studies in constructive direction.

## 5. Conclusion

In the present study, promising phytochemicals were identified from *Bacopa monnieri* and *Mucuna pruriens* against CST as part of substrate reduction therapy for metachromatic leukodystrophy by employing a combinatorial computational approach. Virtual screening based binding score along with ADMET analysis aided in the selection of four potential hits- Stigmastenol, alpha-Amyrenyl acetate, beta-Sitosterol and Jujubogenin, with high bioavailability potential in the brain without any significant toxicity. Furthermore, binding pose and interaction analysis of docked file and simulation trajectory analysis ensured the stability of compounds in the active site. Overall, Stigmastenol is found to be the most suitable drug candidates for CST and alpha-Amyrenyl acetate can be considered as the second best. Overall, due to the satisfactory performance of the other two hits in terms of binding affinity, ADME, toxicity, and stability analysis, the present study recommends *in vitro* inhibition studies using enzymatic screening and cell line-based screening and *in vivo* analysis of all the four compounds. Based on the performance of the selected compounds, the lead optimization would be the future course of action using QSAR and a computer-aided drug design approach.

## Supporting information

**S1 File.**
(DOCX)

## Author Contributions

**Conceptualization:** Nivedita Singh.

**Data curation:** Nivedita Singh.

**Formal analysis:** Nivedita Singh.

**Funding acquisition:** Nivedita Singh.

**Investigation:** Nivedita Singh.

**Methodology:** Nivedita Singh.

**Project administration:** Nivedita Singh.

**Resources:** Nivedita Singh.

**Software:** Nivedita Singh.

**Supervision:** Anil Kumar Singh.

**Validation:** Nivedita Singh.

**Visualization:** Nivedita Singh.

**Writing – original draft:** Nivedita Singh.

**Writing – review & editing:** Nivedita Singh, Anil Kumar Singh.

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
