## [Decision Letter · Decision Letter 0]

10 Jun 2024

PONE-D-24-19810Screening of phytoconstituents from Bacopa monnieri (L.) Pennell and Mucuna pruriens (L.) DC. to identify potential inhibitors against Cerebroside Sulfotransferase.PLOS ONE

Dear Dr. SINGH,

Thank you for submitting your manuscript to PLOS ONE. After careful consideration, we feel that it has merit but does not fully meet PLOS ONE’s publication criteria as it currently stands. Therefore, we invite you to submit a revised version of the manuscript that addresses the points raised during the review process.

We look forward to receiving your revised manuscript.

Kind regards,

Rajesh Kumar Pathak, Ph.D.

Academic Editor

PLOS ONE

Journal Requirements:

"Institute of Eminence, Banaras Hindu University, Government of India

 Grant ID: R/Dev/G/6031/IoE/MPDFs/61698"

3. Please note that funding information should not appear in the Acknowledgments section or other areas of your manuscript. We will only publish funding information present in the Funding Statement section of the online submission form. Please remove any funding-related text from the manuscript. 

4. We note that Figure 1 in your submission contain copyrighted images. All PLOS content is published under the Creative Commons Attribution License (CC BY 4.0), which means that the manuscript, images, and Supporting Information files will be freely available online, and any third party is permitted to access, download, copy, distribute, and use these materials in any way, even commercially, with proper attribution. For more information, see our copyright guidelines: http://journals.plos.org/plosone/s/licenses-and-copyright.

1) You may seek permission from the original copyright holder of Figure 1 to publish the content specifically under the CC BY 4.0 license. 

2) If you are unable to obtain permission from the original copyright holder to publish these figures under the CC BY 4.0 license or if the copyright holder’s requirements are incompatible with the CC BY 4.0 license, please either i) remove the figure or ii) supply a replacement figure that complies with the CC BY 4.0 license. Please check copyright information on all replacement figures and update the figure caption with source information. 

If applicable, please specify in the figure caption text when a figure is similar but not identical to the original image and is therefore for illustrative purposes only.

**Additional Editor Comments:**

The manuscript entitled “Screening of phytoconstituents from Bacopa monnieri (L.) Pennell and Mucuna pruriens (L.) DC. to identify potential inhibitors against Cerebroside Sulfotransferase” has undergone extensive review. Based on the reviewers' comments and suggestions, the authors need to revise the manuscript thoroughly to enhance its quality and readability.

Reviewers' comments:

Reviewer's Responses to Questions

**Comments to the Author**

1. Is the manuscript technically sound, and do the data support the conclusions?

Reviewer #1: Yes

Reviewer #2: Yes

Reviewer #3: Yes

2. Has the statistical analysis been performed appropriately and rigorously? 

Reviewer #1: N/A

Reviewer #2: Yes

Reviewer #3: N/A

3. Have the authors made all data underlying the findings in their manuscript fully available?

Reviewer #1: Yes

Reviewer #2: Yes

Reviewer #3: Yes

4. Is the manuscript presented in an intelligible fashion and written in standard English?

Reviewer #1: Yes

Reviewer #2: Yes

Reviewer #3: Yes

5. Review Comments to the Author

Reviewer #1: Some points for revision are as follows;

1. Check all full stop placed before and after reference citation.

2. Better to write all amino acids with their position as Lys82 rather than LYS82.

3. Check all abbreviations such as TPSA, HBA, HBD, and others not defined.

4. What is meaning of Predicted toxicity class 1-6? (which one is good, better, best, or worst?)

5. In table 3, column interaction type and distance not represented a well and meaningful way. Instead of these column, add a column H-bonding residues.

6. In abstract, sentence "These compounds bound to the active site pocket of CST by interacting with LYS82, LYS85, SER89, TYR176, PHE170, PHE177 in the binding pocket. " say something but residues in table 3 indicate something different. Check and revise.

7. Check capital and small case. use an uniform style. case: upper panel Lower panel

8. Check this sentence "The average intramolecular hydrogen bonds for free CST, CST-GC, IMPHY001534, CST-IMPHY004141, CST-IMPHY009537, and IMPHY014836 were 175.37, 186.13, 172.20, 180.62, 174.09, and 175.8, respectively"

9. It will be more better to use compound name rather than ID as there exists different ID for the same compound in different chemical databases.

10. In table 5, interaction type and distance not represented in meaningful way. Instead of these column, add a column H-bonding residues.

11. There are many more figures, and many subfigures placed on a figure. I suggest you to place some figures as supplementary. Also, improve the quality of text and clarity of figures.

Reviewer #2: The paper titled "Screening of phytoconstituents from Bacopa monnieri (L.) Pennell and Mucuna pruriens (L.) DC. to identify potential inhibitors against Cerebroside Sulfotransferase" presents a comprehensive computational study aimed at identifying potential inhibitors against Cerebroside Sulfotransferase (CST) as a therapeutic target for metachromatic leukodystrophy (MLD). The study employs a multistep virtual screening approach, molecular dynamics simulations, and pharmacokinetic analyses to identify four phytoconstituents with potential inhibitory activity against CST. The paper is well-structured and provides detailed insights into each step of the computational methodology, along with thorough results and discussion sections.

The introduction provides a thorough overview of MLD, its pathophysiology, and current therapeutic approaches, which contextualizes the significance of identifying CST inhibitors. However, it would be beneficial to provide more context on the existing challenges and limitations in developing CST inhibitors, particularly focusing on the lack of experimental structural data for CST, which necessitates the use of computational modeling and virtual screening approaches.

The methodology section is comprehensive and well-described. It provides sufficient details on protein preparation, ligand screening, molecular docking, pharmacokinetic analyses, MM-PB/GBSA calculations, molecular dynamics simulations, and cross-target identification. However, it would be helpful to include the programming script used for each step of the computational analysis, as this can impact reproducibility and compatibility with future studies.

The results section presents a detailed analysis of the screening process, including the identification of top hits based on binding scores, pharmacokinetic properties, MM-PB/GBSA calculations, molecular dynamics simulations, and cross-target identification.

The discussion section effectively interprets the findings in the context of the study objectives and provides insights into the potential implications of the identified inhibitors for MLD therapy. However, it would be valuable to discuss the limitations of the computational approach employed, including assumptions and uncertainties associated with virtual screening, molecular dynamics simulations, and pharmacokinetic predictions.

Additionally, discussing the potential experimental validation of the identified inhibitors, such as in vitro enzymatic assays or cell-based assays, would strengthen the discussion and provide direction for future research.

The conclusion provides a concise summary of the study findings and emphasizes the potential of the identified phytoconstituents as CST inhibitors for MLD therapy. It might be beneficial to include recommendations for future research directions, such as experimental validation studies or further optimization of the identified inhibitors.

Ensure the manuscript is free of typographical and grammatical errors. Proofreading and editing will enhance readability.

Overall, the paper presents a rigorous computational study with well-defined methodologies and comprehensive analyses. The findings contribute to the understanding of CST inhibition for MLD therapy and offer potential drug candidates for further investigation.

Addressing minor points such as providing additional context, discussing limitations, and suggesting future research directions would enhance the clarity and impact of the paper. I recommend accepting the manuscript after these minor revisions are made.

Reviewer #3: The present paper describes screening of phytoconstituents from Bacopa monnieri (L.) Pennell and Mucuna

pruriens (L.) DC. to identify potential inhibitors against Cerebroside Sulfotransferase. The research is well documented and presented through docking analysis and they have identified IMPHY009537 from Bacopa monnieri IMPHY004141 from Mucuna pruriens as the second-best performing inhibitor against CST. The studies are important for development of oral drug for inhibiting CST and to inhibit the metachromatic leukodystrophy disease.

6. PLOS authors have the option to publish the peer review history of their article (what does this mean?). If published, this will include your full peer review and any attached files.

Reviewer #1: No

Reviewer #2: **Yes: **Sutanu Nandi

Reviewer #3: **Yes: **Dr Dinesh Pandey

---

## [Author Response · Author response to Decision Letter 0]

17 Jun 2024

We have addressed all the comments of editor and reviewer in separate file.

---

## [Decision Letter · Decision Letter 1]

4 Jul 2024

Screening of phytoconstituents from Bacopa monnieri (L.) Pennell and Mucuna pruriens (L.) DC. to identify potential inhibitors against Cerebroside Sulfotransferase.

PONE-D-24-19810R1

Dear Dr. SINGH,

We’re pleased to inform you that your manuscript has been judged scientifically suitable for publication and will be formally accepted for publication once it meets all outstanding technical requirements.

Kind regards,

Rajesh Kumar Pathak, Ph.D.

Academic Editor

PLOS ONE

Additional Editor Comments (optional):

The authors have satisfactorily addressed all the comments and concerns raised by the reviewers. Therefore, I recommend the manuscript for publication.

Reviewers' comments:

Reviewer's Responses to Questions

**Comments to the Author**

1. If the authors have adequately addressed your comments raised in a previous round of review and you feel that this manuscript is now acceptable for publication, you may indicate that here to bypass the “Comments to the Author” section, enter your conflict of interest statement in the “Confidential to Editor” section, and submit your "Accept" recommendation.

Reviewer #1: (No Response)

Reviewer #2: All comments have been addressed

2. Is the manuscript technically sound, and do the data support the conclusions?

Reviewer #1: Yes

Reviewer #2: Yes

3. Has the statistical analysis been performed appropriately and rigorously? 

Reviewer #1: N/A

Reviewer #2: N/A

4. Have the authors made all data underlying the findings in their manuscript fully available?

Reviewer #1: Yes

Reviewer #2: Yes

5. Is the manuscript presented in an intelligible fashion and written in standard English?

Reviewer #1: Yes

Reviewer #2: Yes

6. Review Comments to the Author

Reviewer #1: (No Response)

Reviewer #2: The authors have adequately addressed all of my comments, such as providing additional context, discussing limitations, and suggesting future research directions. These changes enhance the clarity and impact of the paper. I recommend accepting the manuscript.

7. PLOS authors have the option to publish the peer review history of their article (what does this mean?). If published, this will include your full peer review and any attached files.

Reviewer #1: **Yes: **Dev Bukhsh Singh

Reviewer #2: No

---

## [Editor Report · Acceptance letter]

27 Aug 2024

PONE-D-24-19810R1 

PLOS ONE

Dear Dr. Singh, 

I'm pleased to inform you that your manuscript has been deemed suitable for publication in PLOS ONE. Congratulations! Your manuscript is now being handed over to our production team.

Kind regards, 

on behalf of

Dr. Rajesh Kumar Pathak 

Academic Editor

PLOS ONE